# The Wildman Programme—Rehabilitation and Reconnection with Nature for Men with Mental or Physical Health Problems—A Matched-Control Study

**DOI:** 10.3390/ijerph182111465

**Published:** 2021-10-31

**Authors:** Simon Høegmark, Tonny Elmose Andersen, Patrik Grahn, Anna Mejldal, Kirsten K. Roessler

**Affiliations:** 1Department of Psychology, University of Southern Denmark, 5000 Odense, Denmark; tandersen@health.sdu.dk (T.E.A.); kroessler@health.sdu.dk (K.K.R.); 2Department of People and Society, Swedish University of Agricultural Sciences, SE-234 56 Alnarp, Sweden; patrik.grahn@slu.se; 3Unit of Clinical Alcohol Research, Institute of Clinical Research, University of Southern Denmark, 5000 Odense, Denmark; amejldal@health.sdu.dk

**Keywords:** chronic diseases, instoration, long-term illnesses, mental health, NBMC method, nature-based rehabilitation, restorative environments, stress, supportive environments, quality of life

## Abstract

Men with health problems refuse to participate in rehabilitation programmes and drop out of healthcare offerings more often than women. Therefore, a nature-based rehabilitation programme was tailored specific to men with mental health problems, and long-term illnesses. The rehabilitation programme combines the use of nature, body, mind, and community spirit (NBMC) and is called the ‘Wildman Programme’. The presented study was designed as a matched-control study with an intervention group participating in the Wildman Programme (N = 114) compared to a control group receiving treatment as usual (N = 39). Outcomes were measured at baseline (T1), post-intervention (T2), and 6 months post-intervention (T3). The primary outcome was the participants’ quality of life measured by WHOQOL-BREF, which consists of four domains: physical health, psychological health, social relationships, and environment. The secondary outcomes were the level of stress measured by the Perceived Stress Scale (PSS), and the participants’ emotional experience in relation to nature, measured by the Perceived Restorativeness Scale (PRS). The intervention group improved significantly in the physical and psychological WHOQOL-BREF domains and in PSS at both follow-ups. The participants’ interest in using nature for restoration increased significantly as well. The only detectable difference between the control group and the intervention group was in the WHOQOL-BREF physical domain at the 6-month follow-up. For further studies, we recommend testing the effect of the Wildman Programme in an RCT study.

## 1. Introduction

According to the World Health Organization (WHO), stress and mental illnesses have become widespread public health problems and today are to be described as pandemics with great human and societal costs [1]. These problems are reflected in the Danish population as well, and the proportion of people in Denmark experiencing a high level of distress has increased from 20.8% to 25.1% between 2010 and 2017 [2].

While mental health problems can affect people’s physical health, somatic illnesses in turn can lead to psychological consequences such as stress, anxiety, or depression [1,2,3]. The incidence of long-term illness and chronic disease is increasing and at least one-third of the Danish population above 16 years of age suffer from one or more long-term physical or mental illnesses [2].

Although more Danish women than men suffer from a long-term illness, men die 4.1 years earlier [4], and research shows that they are less prone to seek help when becoming ill. It also appears that women are more satisfied with the offerings available in the public healthcare system [4,5], while the dropout rate of men in traditional rehabilitation programmes is high [5,6,7,8].

### 1.1. Nature as Treatment

In the last 40 years, research in nature-based interventions as treatment and health promotion have been carried out, and several systematic reviews have found that stays in nature can promote health and well-being [9,10,11]. Furthermore, research in nature-based interventions show positive effects in relation to stress-related mental illnesses [12,13,14,15,16,17,18].

Psychologists and researchers Stephen Kaplan and Rachel Kaplan from the United States took inspiration from Edward Wilsons’ [15] Biophilia theory, as well as from the psychologist William James’ studies on voluntary and involuntary attention [17] to develop the Attention Restoration Theory (ART) [19,20]. It describes how staying in nature can help us rest our minds because we primarily use our attention in a soft and open way that has a healing effect on us. The ART focuses on how we perceive and react to our surrounding environments, and Kaplan and Kaplan describe how we distinguish between two basic forms of attention: (1) directed attention and (2) soft fascination. Directed attention is more demanding for us, while soft fascination is a state of attention where the parasympathetic nervous system is activated, and the body and mind can begin to heal [20]. Kaplan and Kaplan suggest four qualities that must be present for nature to have a healing or restorative effect: (1) being away—offering a feeling of being away from everyday life; (2) extent—giving a feeling of an uninterrupted cohesive world in itself; (3) fascination—offering opportunities for fascination; and (4) compatible—being in accordance with individual needs and abilities [19].

The Stress Reduction Theory (SRT), designed by Professor Roger Ulrich, is another frequently referenced theory [15,16,21,22,23]. He claims that man has an innate readiness to quickly understand both dangers and safe environments in nature. Stressed people can therefore quickly recover in natural environments that have stress-reducing properties, such as being bright and open, with relatively sparse stands of large older trees and preferably overlooking water. The Swedish landscape architect and professor Patrik Grahn developed the Supportive Environment Theory (SET) [24,25]. The SET is based on a combination of preferred natural environments and a positive and restorative impact on our health. The SET describes our need for supportive environments and how people need three different types of supportive environments: (1) physical (which can be divided into different so-called perceived sensory dimensions), (2) social (friends, colleagues, family, neighborhood) and (3) cultural (all types of activities that fill everyday life). In SET, the qualities a natural environment must have to have a restorative effect on our body and mind are defined. Through several studies, Grahn and Stigsdotter have identified eight qualities, also called perceived sensory dimensions (PSD), that people especially prefer when staying in natural environments. These qualities have specific restorative and supportive functions [24].

The above shows that research over the past four decades has proven that staying in and/or exposure to natural environments is health-promotive, e.g., reduces high stress levels and provides recovery from directed attention fatigue. Furthermore, research shows that nature-based therapy works for certain diagnoses, e.g., for people with stress-related mental illness [15].

Based on this, and on the above three theories, a nature-based therapy program, especially focused on men with long-term ill health, has been created: the Wildman Programme. The background and thoughts behind this programme have been described in more detail in [25].

### 1.2. The Wildman Programme

The health problems among Danish men and their resistance towards existing health offers called for a new kind of rehabilitation program. Therefore, the ‘Wildman Programme’ was developed as a nature-based rehabilitation for men. Building on theories and research within nature-based treatment and rehabilitation, it is our belief that a disconnection from nature is one of the reasons for health problems and rebuilding connection with nature is therefore considered to be important for people’s well-being [26,27]. We also assume that some men might find it easier to join a rehabilitation programme in natural surroundings than in a more clinical indoor settings [28]. Our intention was to find out if a nature-based rehabilitation course could broaden the range of Danish rehabilitation programmes appealing to men.

The course was named the ‘Wildman Programme’ and the purpose of the programme was to improve quality of life and reduce symptoms of stress among the participating men [25,26,27,28,29,30,31]. The Wildman Programme was theoretically based on four pillars: nature, body, mind, and community spirit, and the method was called NBMC [25].

The target of the Wildman Programme was to help participants to get a break from everyday life and to reconnect with nature by using safe nature environments. The programme was practiced in five local settings in rural areas with different nature qualities, selected for their restorative and supportive properties [24,27,32] The approach of the programme was inspired by the nature-based theories mentioned above in combination with psycho-evolutionary theory, nature guidance, meditation, and Qigong [33,34]. Qigong is a Chinese system of coordinated movements and body-postures combined with meditations and focuses on cultivating and balancing the flow of life energy in the body [33,34].

The activities within the four pillars of the programme consisted of (1) nature—presentation of scenic areas, sensory activities, silent walking, and fascinating stories about nature and how we are connected to the larger circle of life; (2) body awareness training— breathing exercises, outdoor playing, balance training, Qigong, and other kinds of physical activities; (3) mind relaxation and attention training—walking, standing, sitting and meditating while lying down, outdoor sittings, and narrative meditations; and (4) supporting community spirit—bonfire cooking, talks, and storytelling.

While building on existing knowledge and research within nature-based treatments, the Wildman Programme represents a new form of rehabilitation programme targeted especially at men, practiced in local natural settings, and designed for a relatively large group of men (10–20 participants) being heterogeneous in terms of health problems and diseases. The method in the Wildman Programme has been developed in a pilot project [30,31]. The theoretical framework and the research design are described in more detail elsewhere [25].

### 1.3. Aim

The aim of this study was to examine whether the nature-based Wildman Programme and the NBMC method could improve quality of life and reduce symptoms of stress among men in a heterogeneous group with mental health problems or long-term illnesses. Furthermore, the participants’ experience of restoration in natural environments was examined.

Based on results from the pilot study [30,31], we hypothesised that the study would show that the Wildman Programme has a positive effect on the participating men’s quality of life and symptoms of stress, and that it would appeal to the target group of men.

## 2. Materials and Methods

### 2.1. Study Design

The study was designed as a matched-control study. Outcomes were measured at baseline (T1), post-intervention (T2), and six months post-intervention (T3). The study included an intervention group of men participating in the Wildman Programme and a control group of men participating in treatment as usual (TAU), offered by the local healthcare centre, e.g., physiotherapy, relaxation, rehabilitation, and mindfulness. The control group did not receive any other forms of nature-based intervention during the study. The data collection of this study took place during the period of February 2018–March 2021. The last two years of the study were marked by the COVID-19 pandemic and the social restrictions.

### 2.2. Participants

The participants in both groups were men having mental health problems or a long-term illness recruited by (1) the local healthcare centre, (2) job centre, and (3) through their general practitioner (GP). All participants in the study lived in the municipalities of Svendborg or Faaborg-Midtfyn in Denmark. They suffered from stress (ICD: F43.8 and F43.9), anxiety (ICD: F41.2), or depression (ICD: F32.0) according to the International Classification of Diseases from the American Medical Association, (ICD-10-CM) [35] and/or from the following long-term illnesses: diabetes type-2 (ICD: E11), cancer (ICD: C80.1), post-cancer (ICD: Z08), heart disease (ICD I51.9), chronic obstructive pulmonary disease (COPD) (ICD: J44.9), or pain (ICD R52). The men who participated in the intervention group were not interested in participating in traditional rehabilitation offers and had not sufficiently benefitted from the existing rehabilitation offers in the health centres.

During the recruitment process, a close collaboration was established with the local healthcare centres and the job centres in the two municipalities, and an information campaign was carried out for the local general practitioners (GPs), so they could refer men to this study. In each of the two municipalities, a project manager was hired who became responsible for recruiting participants for the intervention group and the control group, as well as for handing out and collecting the questionnaires to both groups. The control group was selected based on the same criteria as the intervention group. The recruitment of men to the control group was carried out by the two project managers in the healthcare centres, who also recruited the participants for the intervention group.

The referred men to the intervention group were invited to an interview before they were included in the Wildman Programme, and it was decided whether they were ready to start both in terms of motivation and practical conditions such as current treatment. Participation in the Wildman Programme was voluntary and the University of Southern Denmark Research & Innovation Organization, SDU RIO, ethically approved the study. (ClinicalTrials.gov. NCT04073524.) (accessed 29 August 2019).

### 2.3. Venue

The Wildman Programme took place over nine weeks in different natural environments. The selected areas where chosen based on variation in nature types, their natural qualities, and sites to infuse a feeling of being away from everyday life. It was also important that the areas were accessible by public transport and not more than a 20-min drive away from the city, allowing the participants to visit and use the nature sites during and after the course was completed alone or together with friends and family. The Wildman Programme took place all year round and in all kinds of weather.

The participants in the Wildman Programme met five times at a base camp placed at a nature school with a campfire hut surrounded by a forest, hilly landscape, and close to the shore. The other four times, the Wildman Programme took place in various selected natural environments in the southern part of Funen in Denmark. The areas chosen were a forest, a tunnel valley with a stream, a hilly landscape, an open landscape with a meadow area, and the sea and a beach close to a forest (Figure 1). The natural areas that formed the Wildman Programme were typical for the landscape in Denmark and easy to access for the participants regardless of the resources available.

### 2.4. Intervention

The programme lasted nine weeks, where the group met once a week for three hours. Two professionals led the program: a health professional, e.g., a physiotherapist, a nurse, or a psychologist, and a nature guide, who had experience with the target group. Both group leaders were trained in the NBMC method of the Wildman Programme.

During the nine-week course, the participants were introduced to exercises they could practice at home. The home activities consisted of breathing exercises and adapted Qigong exercises for 15 min of training a day. In addition, the participants were encouraged to find their own personal supportive nature environment in their local area during the course.

The intervention during the Wildman Programme was structured, and the intensity was built up during the course. There were fixed elements that were repeated every time the group met, supplemented by new activities from time to time. The activities were constantly adapted to the seasons and the weather and level of function in the group. The four pillars in the NBMC method in the Wildman Programme are described below.

#### 2.4.1. Nature

The participants were introduced to different nature experiences and stories about plants, trees, and animals with the purpose of opening up their awareness towards nature and to trigger their fascination [18]. They were also introduced to fishing, gathering plants, and cooking meals on an open fire [25,36,37,38,39]. By gaining greater insight, skills, and knowledge, dimensions and experiences in nature could be opened up and allowed for close connections to be built [19,27,32]. The participants in the Wildman Programme also experienced the elements of nature and the cycle of nature with changing weather conditions and changing seasons [25,30,31].

#### 2.4.2. Body

Exercises to strengthen body awareness, such as sensory exercises, Qigong, balance training, and plays, were an essential focus in the course [32]. Simple Qigong exercises were introduced, consisting of whole-body movements practiced in a flow and adjusted to the participants’ level of function.

Qigong exercises strengthen body awareness, balance, energy flow, and flexibility all at the same time [33,34,40,41]. Variations in the terrain of the natural environments of the course were used for group silent walks outside walking paths, which can stimulate the senses of the participants, their circuit, and their balance. Also, sensory exercises [42] were a central part of the Wildman Programme. Sensory exercises were practised by listening to, looking at, tasting, touching, and smelling nature and by isolating one sense at a time and then combining them. Sensory exercises were also included in meditations, body scanning, quiet walks, short quiet sitting quests, and plays.

All the exercises helped the participants move from their head and down into their body [42] to feel the presence of the moment more frequently. By moving in many ways and by training the senses, the body map can become more efficient and accurate, and the participants may find it easier to learn new things, find their own resources, and thus become more confident in their abilities.

#### 2.4.3. Mind

An important purpose of the Wildman Programme was to allow the participants’ minds a break from thoughts and worries and to activate the parasympathetic nervous system to let restorative processes happen [43].

To promote their mental health, the participants were introduced to various mindfulness-inspired exercises, outdoor sittings, narrative meditations, and different kinds of attention training [24,32,40,44,45,46,47,48]. Symbols, metaphors, and visualisation in nature were used to support the participants in finding more inner calmness and hope for change in a difficult life situation [49,50,51,52,53,54,55,56,57].

Outdoor sittings were guided gradually, and simple breathing exercises were introduced to support the participants’ movement from predominantly sympathetic nerve activity to parasympathetic nerve activity. It was assumed that the effect of the breathing exercises could be reinforced in nature [58]

#### 2.4.4. Community Spirit

A strong community spirit among the participating men was crucial in the Wildman Programme. Research shows that a more informal, organic, relaxed, free, and spontaneous atmosphere can arise when you meet up in nature [55,56]. In the Wildman Programme, a relaxed, supportive, open, and positive atmosphere in the group was the goal. There was little focus on health problems and more on individual and common resources within the group. The men could make a new and common story with a new meaning of life and a new understanding of the world [59,60,61,62].

The bonfire was an essential meeting point in the group. The Wildman Programme always started the day by having the group meet around a bonfire, and when the day ended, the group gathered again for bonfire food and bonfire tales as our ancestors have done since they learned how to control fire [62,63,64].

Social support is important in times of life crises and the community spirit [65,66] was a significant part of the course, supported by common exercises, plays, talks, and stories around the bonfire and by bringing the resources of the participants into play; an essential goal was that the unity in the group was maintained after the Wildman Programme. Therefore, an association was established for former course participants, called the ‘Wildman Association’.

On the last day of the Wildman Programme the participants were invited to join one of two local Wildman Associations. The Wildman Associations are volunteer bridge-building offers where the men can continue the social community in nature and together practice the simple nature-based activities after the course had ended.

### 2.5. Outcomes

#### 2.5.1. Primary Outcome

The primary outcome of the study was the participants’ quality of life.

Quality of life was assessed by The World Health Organization’s Quality-of-Life Scale: WHOQOL-BREF [67]. The WHOQOL-BREF consists of 26 items including 2 overall questions about quality of life and health, and the remaining items are divided into four domains:Physical health: 7 items.Psychological health: 6 items.Social relationships: 3 items.Environment: 8 items.

All items have a range from 1 to 5. Three items are negatively framed questions and must be reversed. The domain scores are scaled in a positive direction, which means that a high score denotes a high quality of life. The mean of item scores within each domain is calculated and multiplied by four, resulting in the domain score. This converts the domain scores to a range between 4 and 20.

#### 2.5.2. Secondary Outcomes

The secondary outcomes of the study were the participants’ level of stress and self-perceived experience of restitution in nature measured by the following scales:The Perceived Stress Scale (PSS) [68]: PSS examines how different situations affect feelings and perceived stress in daily living within the last month. The scale consists of 10 items in a five-point Likert Scale. The scores range from 0 to 4 for the questions 1, 2, 3, 6, 9, and 10. The scores of the questions 4, 5, 7, and 8 are reversed. The scores for each item are added to get a total. The individual scores on the PSS can range from 0 to 40. Higher scores indicate higher perceived stress. Scores ranging from 0 to 13 are considered as a low level of perceived stress, scores ranging from 14 to 26 are considered as a moderate level of perceived stress, and scores ranging from 27 to 40 are considered as a high level of perceived stress [68].The Perceived Restorativeness Scale (PRS) [69]: PRS measures, in 26 items, four different categories of self-experienced restitution related to spontaneous attention: fascination, being away, extent, and compatibility. The scale is used for measuring meditation practice and attention training in natural environments [70].

Since only the intervention group participated in a nature-based rehabilitation programme, the items of the PRS were only answered by the intervention group and not by the participants in the control group.

### 2.6. Statistical Analysis

The range and distribution of all key socio-demographic and outcome variables at baseline were calculated and compared across the intervention group and the control group using chi-square tests for categorical data and *t*-tests for continuous data. Paired *t*-tests were applied to test differences by group from each follow-up to baseline on all non-missing WHOQOL-BREF domain scores, PSS scores, and PRS scores, and Cohen’s d was calculated to evaluate the effect sizes.

Following this, a linear mixed-model analysis was used to examine the trajectories of the outcomes by group over time. Specifically, the influence of time (baseline, post-intervention (T1), and 6-month follow-up (T2)), group (intervention (Wildman Programme) or control (TAU)), and the interaction between time point and group on outcomes within WHOQOL-BREF domain scores and PSS scores was investigated.

For each of the dependent variables, the need for a subject-specific random intercept as well as a random slope was tested. Assuming the dropout mechanism is missing at random (MAR), linear mixed models deal efficiently with missing values due to dropout using the maximum likelihood estimator. Therefore, with the mixed-effects model approach, all available data were used. The predicted changes from baseline to each follow-up were calculated using the results from the linear mixed models. Using the estimates from these models, the changes between timepoints, called ‘predicted mean differences’, were estimated for each group, and then differences between groups were calculated.

To further account for the non-randomisation, two analyses were performed for each outcome measure: (1) unadjusted, and (2) adjusted for referral type, physical illness, psychological illness, and current treatment as these differed at baseline. All analyses were conducted using Stata version 16.

## 3. Results

### 3.1. Flow of Participants

A total of 153 men were included in the study: 114 in the intervention group, and 39 men in the control group that received treatment as usual (TAU) (see Figure 2).

In the intervention group, 73% (*n* = 83) of the men completed the 9-week follow-up questionnaire, while 50% (*n* = 57) completed the 6-month follow-up. In the control group, 85% (*n* = 33) completed the 9-week follow-up questionnaire, and 76% (*n* = 30) completed the 6-month follow-up.

### 3.2. Socio-Demographic Characteristics

Table 1 shows the socio-demographic characteristics and other variables for the participants in the intervention group and the control group before the intervention.

Table 1 shows the similarities and differences between the intervention group and the control group. The mean age in the intervention group and the control group was around 55 years in both groups. Approximately 50% of both groups of men had not completed a higher education, whereas the other half had completed an intermediate or long higher education. About 25% of the participants in both groups were living alone, and about 80% had one or more children.

The two groups differed in some of the characteristics. In the intervention group, a higher proportion of the participants were unemployed or in job training/specialised courses (flex, resource, rehabilitation) compared with the control group (intervention 29.9% vs. control 10.5%). The two groups differed in relation to referral type. While both groups were mainly referred to the study from other sources, the intervention group had a higher share of participants referred from the job centre, while a higher proportion of the men in the control group had been referred from their general practitioners. Furthermore, the two groups differed in having current physical illnesses (intervention 57% vs. control 76%) and current psychological illnesses (intervention 53% vs. control 29%). Also, participants in the control group were more likely to currently be in treatment for their illnesses (intervention 63% vs. control 89%). Finally, the intervention group were more likely to have current contact with a psychiatric hospital ward (intervention 22% vs. control 3%).

### 3.3. Results of the Statistical Analysis

Table 2 show the results of the statistical analysis of the primary and secondary outcomes for the intervention group and the control group.

In Table 2, mean values for the primary and secondary outcomes at baseline and T1 and T2 are shown. Mean values were computed on those who had a non-missing value at the time point in question. Furthermore, mean differences between baseline and each of the follow-ups, together with *p*-values from the paired *t*-tests and Cohen’s d values, are shown, calculated on those present for both baseline and 9-week follow-up, and baseline and 6-month follow-up, respectively.

Table 2 shows that the participants in the intervention group completing the 9-week follow-up improved significantly in the physical WHOQOL-BREF domain, increasing by 0.84 (SD 2.18, *p* = 0.008), and in the psychological WHOQOL-BREF domain, increasing by 0.46 (SD 1.87, *p* = 0.029). Significant improvements were also shown in PSS with a reduction in stress symptoms by 3.63 (SD 5.13, *p* < 0.001) and in PRS, increasing by 18.57 (SD 27.14, *p* < 0.001). Cohen’s d showed a small effect size in the physical and psychological WHOQOL-BREF domains, and medium effect sizes in PSS and PRS.

A similar outcome presented for the participants in the intervention group completing the 6-month follow-up, with significant improvements in the physical and psychological WHOQOL-BREF domains with slightly higher effect sizes, and in PSS and in PRS, though with lower effect sizes. Furthermore, at the 6-month follow-up, the social WHOQOL domain also showed a significant improvement by 0.90 (SD 2.91, *p* = 0.024).

The control group only experienced a significant change in PSS among the men completing the 9-week follow-up, with a 2.75 reduction in their stress symptoms (SD 4.54, *p* = 0.0018) with a medium effect size. There were no significant changes from baseline detected among the participants in the control group completing the 6-month follow-up.

Table 3 and Figure 2 and Figure 3 show the results of the intention-to-treat analysis with linear mixed models and the predicted values and predicted mean differences of primary and secondary outcomes over time and by group, from adjusted linear mixed models.

Examining the adjusted analysis, the intervention group improved significantly in the WHOQOL-BREF physical domain at both follow-ups (9 weeks: PMD (SE) 0.97 (0.25), *p* < 0.001; 6 months: PMD (SE) 1.54 (0.33), *p* < 0.001) and in the WHOQOL-BREF psychological domain at both follow-ups (9 weeks: PMD (SE) 0.49 (0.22), *p* = 0.0236); 6-month follow-up (PMD (SE) 0.95 (0.35), *p* = 0.0072). Furthermore, significant improvements were found at both follow-ups for the PSS (9-week PMD (SE) −3.88 (0.60), *p* < 0.001; 6-month PMD (SE) −4.63 (0.94), *p* < 0.001) and PRS (9-week PMD (SE) 17.76 (3.48), *p* < 0.001; 6-month PMD (SE) 16.12 (3.99), *p* < 0.001).

In the control group, significant improvement was found only in the WHOQOL environmental domain at the 9-week follow-up (PMD (SE) 0.57 (0.28); *p* = 0.044).

However, the only detectable difference in PMD between the intervention group and the control group was in the WHOQOL-BREF physical domain for the PMD at the 6-month follow-up (difference in PMD (SE) 1.42 (0.51); *p* = 0.0056).

Figure 3 and Figure 4 show the adjusted analyses of the differences in the two groups in WHOQOL-BREF, PSS, and PRS.

## 4. Discussion

In this study, our aim was to examine whether the nature-based Wildman Programme using the NBMC method could improve quality of life and reduce symptoms of stress among men in a heterogeneous group with mental health problems or long-term illnesses. Furthermore, the participants’ experience of restoration in natural environments was examined.

### 4.1. Effects of the Wildman Programme

The results of this study showed that the men participating in the Wildman Programme improved significantly regarding their physical and psychological quality of life domains in WHOQOL-BREF both after nine weeks and six months, and they also improved significantly on perceived stress measured by PSS at both times of follow-up. The social quality of life domain in WHOQOL-BREF showed significant improvements at the six-month follow-up in Table 2 (including only completers) as well. However, the social domain was not significant for the intervention group in the analysis with the adjusted linear mixed models using all available data. The study showed a significant increase in the participants’ perception of and interest in nature and in their use of nature as a restorative environment measured by PRS.

Significant improvements on the primary and secondary outcomes were not found for the men in the control group receiving treatment as usual, except for a significant improvement in the participants’ environmental health domain in WHOQOL-BREF at the 9-week follow up.

Contrary to our a priori hypothesis, we did not find that participants in the Wildman Programme improved significantly more on all outcomes over time compared to the participants in the control group who received traditional treatment. A significant difference between the two groups was only found on the physical quality of life domain.

However, the results show that the men in the intervention group had a poorer state of health at the starting position (physical, psychological, and symptoms of stress) based on Figure 3 and Figure 4, and the improvements in these health outcomes were nevertheless at least equally positive as in the control group, and the development was significantly better in terms of the physical WHOQOL-BREF domain.

The study did not manage to ensure a sufficient match between the intervention group and the control group in relation to health variables and the sociodemographic background variables. Therefore, the two groups of men were not found to be sufficiently comparable. One of the reasons for this was that the two healthcare centres had difficulties finding a matching control group to the participants in the Wildman Programme. The group of participants in the Wildman Programme was very broad and included men with a wide range of diagnoses, while the participants in the control group were recruited from different health offerings targeting more narrow diagnostic groups. It is also to be expected that the men who agreed to participate in the control group and answered the questionnaires at both baselines, the 9-week follow-up and the 6 months follow-up, overall had more personal resources and were in better health than the men in the intervention group. This is confirmed by the statistical data showing that the intervention group were worse off on all health outcomes at the beginning of the study (in both physical and psychological WHOQOL-BREF domains and in PSS) compared to the control group.

Furthermore, the two last two years of the data collection of this study (2020–2021) were marked by the COVID-19 pandemic and many restrictions which may have resulted in a smaller control group than desired.

If the base of recruitment had been larger, it would have been optimal with a randomisation of the groups in the research design based on waiting groups to participate in the Wildman Programme.

A sufficient number of participants were included in the intervention group of this study to show a positive development in some of the health outcomes. However, it was not possible to tell if the outcomes were better than the traditional treatment received by the control group, except for the participants’ physical health, where a significant difference was found between the two groups.

The positive effect among the participants in the Wildman Programme showed that the nature-based rehabilitation programme and the NBMC method can improve physical and mental health and reduce symptoms of stress among men with mental health problem or long-term illnesses or chronic diseases. Thus, the study showed that the Wildman Programme and the NBMC method can improve health for men together in a heterogenous group consisting of a wide range of mental and physical illnesses. The health effects were seen to be improved both during the course (after 9 weeks), and 6 months after the course was completed.

The positive effects after 6 months indicate that the Wildman Programme leaves the men participating in the course with tools and a new view of nature that they can also use and benefit from after the course. The results of the PRS confirm this assumption, since a significant improvement was seen in the participants’ experience and perception of nature as an environment for restoration. Moreover, it is an advantage for the anchoring of new habits in the participants’ everyday life and lifestyle that nature is a free and easily accessible resource. The continuing improvements in the participants’ health may also be due to the possibility of continuing the activities together with other former participants in the Wildman Associations. Participating in a binding network doing nature-based health-promoting activities together can make it easier to maintain new habits.

Community spirit is a central focus in the Wildman Programme, and it was a surprise that the social domain in WHOQOL-BREF showed significant improvements only in the paired *t*-test (Table 2) after the 6-month follow-up and not after the 9-week follow-up. However, the adjusted linear mixed models did not show significant improvements at any of the times for follow-up. The improvement seen after the 6-months follow-up in Table 2 can probably be explained by the significant number of men who have joined the two Wildman Associations. This could point to the need for more bridge-building activities from public rehabilitation programmes to local communities and associations where men can meet up with like-minded men to build and maintain relations in the long term.

### 4.2. Appeal to Men

The nature-based Wildman Programme and the NBMC method have been implemented in two municipalities. Initially, the recruitment of men for the course proved difficult. This was expected, since men to a lesser extent than women are prone to participating in rehabilitation offers [4,7]. However, it turned out to be easier to recruit men from the target group as time went by and more and more men experienced rehabilitation in nature. At the end of the study, a large part of the recruitment of men to the Wildman Programme took place by word of mouth, where previous participants recommended the Wildman Programme to men in their network with similar health problems. In both the municipality of Svendborg and the municipality of Faaborg-Midtfyn, there have been waiting lists for participation in the Wildman Programme. After the Wildman Programme ended, the men could continue the social community and the activities of the course in a local Wildman Association. The interest in volunteer participation has been high, and at the time of the conclusion of this study, the two Wildman Associations had 180 members. This indicates that nature-based rehabilitation and nature-based activities appeal to men.

However, the question remains: Why is nature-based rehabilitation attractive for men?

The group of men who participated in the Wildman Programme varied in terms of education, job status, and profession. Engineers, carpenters, directors, teachers, craftsmen, machinists, chefs, and workers participated in the course, as well as unemployed and retired men. The wide appeal to different kinds of men may be due to the fact that in nature social divides are perceived as less significant and are replaced by a feeling of recognisability in each other [32].

Research shows that natural settings are experienced as informal with a relaxed atmosphere that is difficult to create indoors [71,72,73]. Nature can be experienced by the participants as a neutral place where they are not constantly reminded that they are engaged in a health course, and they are to a lesser extent reminded that they are sick. In the Wildman Programme the participants were also met by a different approach than they may have experienced in other rehabilitation programmes, with a focus on their resources, building a team spirit, and stimulating a fascination of nature. This might have had a positive influence on their self-perception and feelings about their own identity.

Many men experience a change in their perception of identity when they get a severe illness and are at risk of losing their job or being on sick leave from their job for an extended period [4]. They may feel that they lose some of their personal value and that life loses some of its meaning because they are no longer able to contribute to their family and society in the same way as before.

Restoring contact and connection to nature can help recreate meaning in life [73]. The feeling of being part of something bigger can make one’s own problems seem smaller and leave room for self-forgetfulness for a moment. There are many indications for a connection between close contact with nature, inner peace, and life satisfaction [74].

The Wildman Programme differs from many other rehabilitation courses since the course was practiced outdoors and in a heterogeneous group whose participants had different health problems and diagnoses. This meant that the illness and treatment of the individual participant took up little space in the course. Instead, it focused on the experience of being together with like-minded men. This might also have had a positive appeal to the participating men. Although the focus was shifted away from the diagnoses, it still required that the health professional course leader had a great insight and high professionalism within the various health problems and diagnoses that were included in the course. It is crucial that the course leaders are aware of disabilities and emotional reactions that are typical of participants with the health problems and illnesses in question. In addition, there is an important coordinating role in relation to the various co-workers that were involved in the individual participant’s case (e.g., job centre, GP, labour union, and workplace), and it can be a challenge as a course leader of a rehabilitation programme consisting of a wide and heterogeneous group of men [75].

### 4.3. Limitations

Our study has limitations. The control group was not sufficiently matched to the intervention group, which makes the study results less solid. The size of the control group did not have the size we had wished for. The plan of this study was to include 52 participants in the control group, but it was difficult to recruit enough men to the control group and this was further made difficult because of the restrictions in relation to COVID-19. This is a limitation that made a match between the two groups difficult. The two groups differed in relation to employment status, referral type, and health profile, and the answers of the questionnaire showed that the intervention group was worse off on both WHOQOL-BREF and PSS at baseline.

In future research on the Wildman Programme, randomisation of the groups in the research design with waiting groups would be preferable.

The questions about the participants’ ‘job status’ and ‘referred from’ in the questionnaire were not detailed enough, and consequently a high proportion of the men answered ‘other’ to the two questions. Interesting information is therefore lacking in relation to these two issues.

A limitation in this study is that the rehabilitation offers received by the control group were not sufficiently covered and included different kinds of rehabilitation programs (except for nature-based interventions). Therefore, the circumstances for the participants in the control group were diverse.

The collection of data has been a challenge, as many of the participants felt it was overwhelming and difficult for them to answer a large questionnaire battery, and it has been a big workload and difficult process for the project managers in the municipalities to get the participants motivated to fill in the questionnaires. In relation to future similar research projects, alternative methods of data collection may be considered.

COVID-19 played a significant role in the last two years of this study, and there were extended periods of time where the Wildman Programme could not be conducted due to official restrictions. The healthcare centres have been closed as well, which has made it more difficult to keep in touch with the participants. Many of the participants were vulnerable to infections of COVID-19 and were discouraged from attending the courses by their own doctor due to the risk of infection, which created an unstable attendance flow during the pandemic.

Social isolation and social distancing were for many people consequences of the many COVID-19 restrictions. This could have affected the scores on the social domain of this study and may have impacted the results from the adjusted linear mixed models.

The participants of this study were not screened for pandemic related stress, which could have affected some of them during the period of COVID-19 restrictions and may to some extent have impacted the results of this study.

## 5. Conclusions

The hypothesis prior to the study was that the study would show that the Wildman Programme had a positive effect on the quality of life and symptoms of stress among men in a heterogenous group with mental health problems and long-term illness, and that the target group of men would find the rehabilitation programme appealing.

It seems that the Wildman Programme has a wide appeal to men from different social groups and professions. The Wildman Programme improves the participants’ physical and psychological health measured by the quality-of-life scale, WHOQOL-BREF. The participants’ level of stress was reduced as well, and they gained a greater experience of nature as a restorative space that they could use to recover. The Wildman Programme showed positive effects in a heterogenous group of men with different kinds of mental health problems and somatic diagnoses, and the course seems to be a relevant alternative rehabilitation offer for men.

The intervention group and the control group were not sufficiently matched, which made comparisons between the two groups uncertain. This is a weakness in this study. It is recommended to rethink the research design and do more research in the field of nature-based interventions, as the method seems to have an appeal to men, and alternative rehabilitation offers are lacking in the Danish health system.

## Figures and Tables

**Figure 1 ijerph-18-11465-f001:**
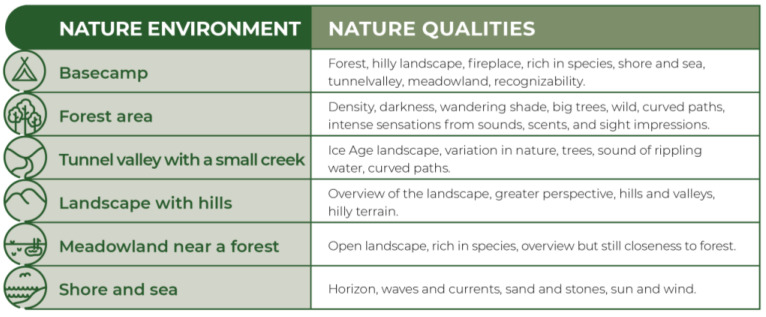
Nature environment.

**Figure 2 ijerph-18-11465-f002:**
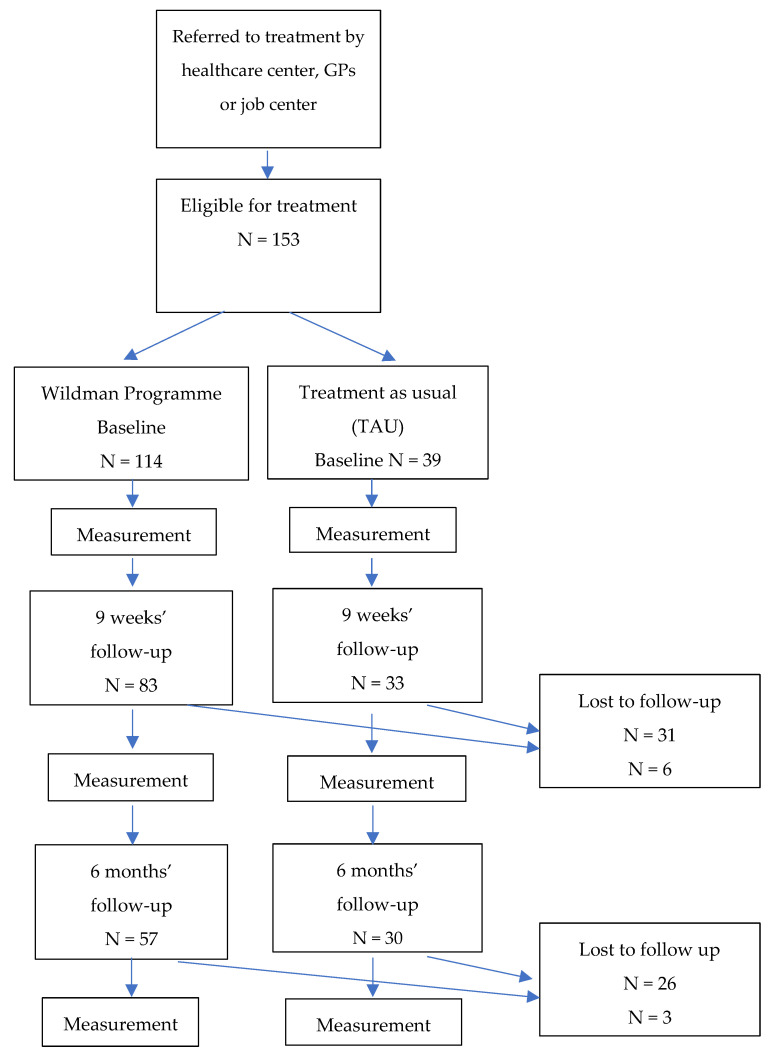
Flowchart for the study of the ‘Wildman Programme’.

**Figure 3 ijerph-18-11465-f003:**
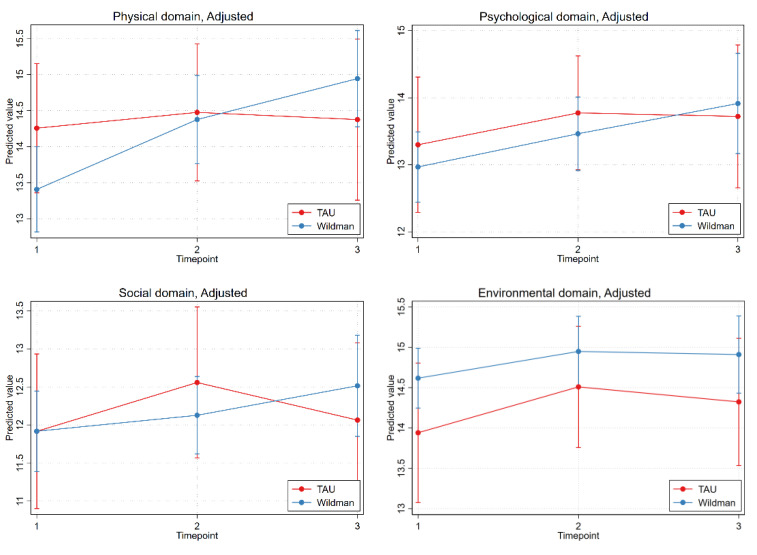
Adjusted analysis of the differences in the development of the intervention group and control group on the four WHOQOL-BREF domains.

**Figure 4 ijerph-18-11465-f004:**
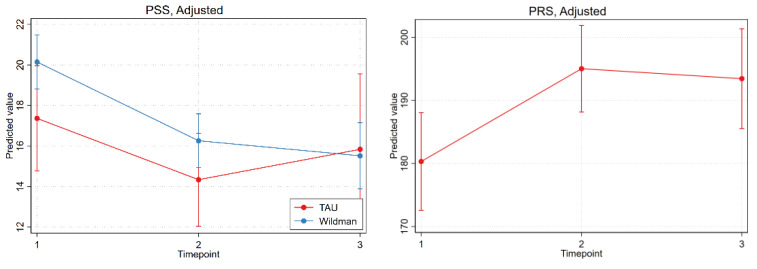
Adjusted analyses and the development in PSS between the participants in the intervention group and the control group and the development in PRS for the intervention group.

**Table 1 ijerph-18-11465-t001:** Sample characteristics at baseline (N = 153).

Sample Characteristics	Control	Intervention	*p*-Value
N	39	114	
Age, mean (SD)	57.55 (10.70) (*n* = 38)	54.60 (13.67) (*n* = 109)	0.23
Educational level (ISCED), *n* (%)			
Lower secondary or less	6 (15.8%)	18 (16.7%)	0.95
Upper secondary	13 (34.2%)	39 (36.1%)	
Short cycle tertiery/bachelor	13 (34.2%)	38 (35.2%)	
Master’s or above	6 (15.8%)	13 (12.0%)	
Currently employed, *n* (%)			
Unemployed	3 (7.9%)	20 (18.7%)	0.13
Employed	13 (34.2%)	21 (19.6%)	
In job training or education	1 (2.6%)	12 (11.2%)	
Retired	12 (31.6%)	28 (26.2%)	
Other	9 (23.7%)	26 (24.3%)	
Cohabiting status, *n* (%)			
Alone	8 (21.1%)	28 (25.7%)	0.57
Cohabiting	30 (78.9%)	81 (74.3%)	
Children, *n* (%)			
Yes	32 (84.2%)	91 (83.5%)	0.92
No	6 (15.8%)	18 (16.5%)	
Referred from, *n* (%)			
General Practitioner (GP)	12 (33.3%)	8 (7.4%)	<0.001
Job centre	3 (8.3%)	38 (35.2%)	
Other	21 (58.3%)	62 (57.4%)	
Physical illness(es), *n* (%)			
Yes	28 (75.7%)	61 (57.0%)	0.044
No	9 (24.3%)	46 (43.0%)	
Psychological illness(es), *n* (%)			
Yes	11 (28.9%)	56 (52.8%)	0.011
No	27 (71.1%)	50 (47.2%)	
In treatment ^1^, *n* (%)			
Yes	32 (88.9%)	65 (63.1%)	0.004
No	4 (11.1%)	38 (36.9%)	
Contact with psychiatric hospital ^2^, *n* (%)			
Yes	1 (3%)	20 (22%)	0.012
No	32 (97%)	70 (78%)	
Medication ^3^, *n* (%)			
Yes	30 (83.3%)	74 (70.5%)	0.13
No	6 (16.7%)	31 (29.5%)	

^1^ Currently in treatment at hospital for their primary illness (e.g., diabetes, cancer, CORP). ^2^ Currently in contact with a psychiatric hospital. ^3^ Currently in medical treatment in relation to their primary illness.

**Table 2 ijerph-18-11465-t002:** Means of primary and secondary outcomes over time in the intervention group (Wildman Programme) and the control group (TAU), paired *t*-tests and effect size measure of difference (Cohen’s d).

Intervention Group (Wildman Programme)									
	Baseline, Mean (SD)	9-Week Follow-Up, Mean (SD)	Difference Baseline to 9-Week Follow-Up, Mean (SD) ^3^	Cohen’s d Baseline to 9-Week Follow-Up ^4^	*p*-Value ^1^	6-Month Follow-Up, Mean (SD)	Difference Baseline to 6-Month Follow-Up, Mean (SD) ^3^	Cohen’s d Baseline to 6-Month Follow-Up ^4^	*p*-Value ^2^
WHOQOL-BREF Physical domain	13.41 (3.05)	14.26 (2.76)	0.84 (2.18)	0.38	0.001	15.15 (2.73)	1.32 (2.66)	0.50	0.000
WHOQOL-BREF Psychological domain	12.67 (3.28)	13.38 (3.02)	0.46 (1.87)	0.24	0.029	14.43 (3.15)	1.10 (2.84)	0.39	0.005
WHOQOL-BREF Social domain	11.74 (2.95)	12.17 (2.53)	0.22 (2.56)	0.09	0.440	12.92 (2.59)	0.90 (2.91)	0.31	0.024
WHOQOL-BREF Environmental domain	14.50 (2.04)	14.96 (2.11)	0.23 (1.59)	0.15	0.191	15.32 (1.91)	0.16 (1.63)	0.10	0.470
PSS	20.38 (7.36)	16.46 (6.11)	−3.63 (5.13)	−0.71	<0.001	14.70 (6.24)	−3.36 (7.31)	−0.46	0.001
PRS	179.26 (35.56)	195.27 (28.11)	18.57 (27.14)	0.68	<0.001	195.37 (30.91)	16.07 (31.41)	0.51	0.002
Control group (TAU)									
	Baseline, mean (SD)	9-week follow-up, mean (SD)	Difference baseline to 9-week follow-up, mean (SD) ^3^	Cohen’s d baseline to 9-week follow-up ^4^	*p*-value ^1^	6-month follow-up, mean (SD)	Difference baseline to 6-month follow-up, mean (SD) ^3^	Cohen’s d baseline to 6-month follow-up ^4^	*p*-value ^2^
WHOQOL Physical domain	14.22 (2.77)	14.61 (2.77)	0.03 (2.41)	0.01	0.951	14.57 (3.37)	0.15 (2.23)	0.07	0.722
WHOQOL Psychological domain	13.86 (3.32)	14.54 (2.65)	0.38 (2.31)	0.16	0.351	14.39 (3.11)	0.41 (2.44)	0.17	0.369
WHOQOL Social domain	12.03 (3.22)	12.83 (2.79)	0.63 (2.58)	0.24	0.180	12.31 (2.96)	0.22 (2.48)	0.09	0.627
WHOQOL Environmental domain	14.05 (2.68)	14.94 (1.91)	0.59 (1.84)	0.32	0.076	14.52 (1.97)	0.29 (1.75)	0.16	0.374
PSS	16.15 (7.49)	13.03 (6.25)	−2.75 (4.54)	−0.61	0.002	14.50 (9.37)	−1.40 (7.83)	−0.18	0.335

^1^*p*-value from paired *t*-test comparing baseline and 9-week follow-up. ^2^*p*-value comparing baseline and 6-month follow-up. ^3^ Differences only calculated on completers (baseline and 9-week/baseline and 6-month, respectively). ^4^ Cohen’s d defined as mean (difference)/SD (difference).

**Table 3 ijerph-18-11465-t003:** Predicted values and predicted mean differences of primary and secondary outcomes over time and by group, from adjusted linear mixed models. ^2^

		Control Group (TAU)			Intervention Group (Wildman Programme)			Intervention vs. Control	
		Predicted Value (SD)	Predicted Mean Difference (SD)	CI (*p*-Value)	Predicted Value (SD)	Predicted Mean Difference (SD)	CI (*p*-Value)	Difference in PMD (SD) ^1^	CI (*p*-Value)
WHOQOL-BREF Physical domain	Baseline	14.26 (0.46)			13.41 (0.30)				
	9-week follow-up	14.48 (0.49)	0.22 (0.46)	−0.69; 1.13 (0.636)	14.38 (0.31)	0.97 (0.25)	0.49; 1.45 (0.0001)	0.75 (0.53)	−0.28; 1.78 (0.1530)
	6-month follow-up	14.38 (0.57)	0.12 (0.39)	−0.65; 0.89 (0.760)	14.95 (0.34)	1.54 (0.33)	0.90; 2.18 (0.0000)	1.42 (0.51)	0.42; 2.42 (0.0056)
WHOQOL-BREF Psychological domain	Baseline	13.30 (0.51)			12.97 (0.27)				
	9-week follow-up	13.77 (0.43)	0.48 (0.43)	0.37; 1.32 (0.2691)	13.46 (0.28)	0.49 (0.22)	0.07; 0.92 (0.0236)	0.02 (0.48)	−0.93; 0.97 (0.9672)
	6-month follow-up	13.72 (0.54)	0.42 (0.48)	−0.51; 1.35 (0.3732)	13.91 (0.38)	0.95 (0.35)	0.26; 1.64 (0.0072)	0.52 (0.59)	−0.63; 1.68 (0.3756)
WHOQOL-BREF Social domain	Baseline	11.92 (0.52)			11.92 (0.27)				
	9-week follow-up	12.56 (0.51)	0.64 (0.47)	−0.27; 1.55 (0.1685)	12.13 (0.26)	0.21 (0.30)	−0.37; 0.79 (0.4794)	−0.43 (0.55)	−1.51; 0.65 (0.4353)
	6-month follow-up	12.06 (0.52)	0.15 (0.41)	−0.65; 0.94 (0.7163)	12.51 (0.34)	0.60 (0.38)	−0.16; 1.35 (0.1203)	0.45 (0.56)	−0.65; 1.55 (0.4239)
WHOQOL-BREF Environmental domain	Baseline	13.94 (0.44)			14.62 (0.19)				
	9-week follow-up	14.51 (0.38)	0.57 (0.28)	0.02; 1.12 (0.0440)	14.95 (0.22)	0.33 (0.19)	−0.04; 0.71 (0.0831)	−0.24 (0.34)	−0.91; 0.43 (0.4875)
	6-month follow-up	14.32 (0.40)	0.38 (0.27)	−0.14; 0.90 (0.1493)	14.91 (0.24)	0.29 (0.22)	−0.15; 0.73 (0.1911)	−0.09 (0.35)	−0.77; 0.59 (0.7969)
PSS	Baseline	17.36 (1.32)			20.14 (0.68)				
	9-week follow-up	14.34 (1.17)	−3.02 (0.93)	−4.84; −1.21 (0.0011)	16.26 (0.68)	−3.88 (0.60)	−5.06; −2.71 (0.0000)	−0.86 (1.10)	−3.02; 1.30 (0.4355)
	6-month follow-up	15.84 (1.90)	−1.53 (1.67)	−4.79; 1.74 (0.3594)	15.52 (0.83)	−4.63 (0.94)	−6.48; −2.78 (0.0000)	−3.10 (1.92)	−6.85; 0.66 (0.1058)
PRS ^3^	Baseline	NA			178.70 (3.90)	Reference		NA	
	9-week follow-up				196.46 (3.13)	17.76 (3.48)	10.95; 24.57 (0.0000)		
	6-month follow-up				194.82 (3.63)	16.12 (3.99)	8.30; 23.94 (0.0001)		

^1^ PMD = Predicted mean difference. ^2^ Linear mixed models adjusted for referral type, physical illness, psychological illness, and current treatment. ^3^ PRS only measured for Wildman group, no adjustments in linear mixed models.

## Data Availability

The data presented in this study are according to Danish law not available for sharing.

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
