# Peer review of "The Wildman Programme—Rehabilitation and Reconnection with Nature for Men with Mental or Physical Health Problems—A Matched-Control Study"

_ijerph, 2021, doi:10.3390/ijerph182111465_

Round 1

Reviewer 1 Report

The study of the Authors aimed to explore whether: the nature-based Wildman Programme and the NBMC method can promote quality of life and reduce symptoms of stress among men in a heterogeneous group with mental health problems or long-term illnesses and chronic diseases; the course can provide the participants with tools to use nature to recover; the Wildman Programme appeals to the target group of men and can encourage more men to participate in a rehabilitation program.

Reading an article of this topic could be beneficial for health professionals. Valuable ascertainments are contained in the article, the manuscript is excellently written.

I recommend an „Accept after minor revision” option for the publication of the article.

There are some parts of the manuscript that I suggest to be rewritten.

Page 1., Lines 38-40.: „These mental health problems are reflected in the Danish population as well, and stress has for the last decade become an increasing public health problem.” Please write distress or effects of stress/distress into this sentence, as stress in itself does not elicit a health problem.

Page 2., Lines 52-55.: it seems that there is a missing part in this sentence.

Figure 2.: in this figure, one participants possible drop-out from the study was not illustrated between the „Eligible for treatment (n=154)” and the boxes of the intervention group and the control group.

FIgure 2.: there is a mistake in this figure where TUA is written instead of TAU.

Page 11-12., Lines 464-470.: „The reason for only 50% of the intervention group responding on the 6-month follow-up questionnaire could be that some of the participants were not used to using digital tools and therefore found it difficult to reply; the length of the questionnaire may have demotivated some of the participants; and the interest in the course and importance of completing the questionnaire may have paled 6 months after the intervention ended. Furthermore, the situation with COVID-19 has limited the contact with former participants since the healthcare centers have been closed for a period.” It would be illustrative if the Authors described the exact start date of the first intervention group and the date of the last 6-month follow-up examination of the project until the submission the manuscript.

Page 13., Lines 482-485.: „Approximately 50% of both groups of men had not completed a higher education, whereas the other half were relatively well-educated, since most of them had completed an intermediate or long higher education after finishing elementary school or high school.” This sentence seems to be a little judgmental, I think it would sound better to write it for example this way: Approximately 50% of both groups of men had not completed a higher education, whereas the other half had completed an intermediate or long higher education after finishing elementary school or high school.

Page 15., Lines 520-522.: „Significant improvements were also shown on PSS with a reducing I stress symptoms by 3.63 (SD 5.13, p<0.001) and on PRS, increasing by 18.57 (SD 27.14, p<0.001).” I think it would be a possible way to correct this sentence by modifying its middle part like this:„reduction in stress symptoms”.

Author Response

October 15, 2021

Thank you very much for your constructive comments and suggestions of improvements of our article. We hereby resubmit a revised edition of the article. Besides the changed made based on the comments of the reviewers, we have ourselves made larger reductions and changes in the text of the manuscript.

Page 1., Lines 38-40.: „These mental health problems are reflected in the Danish population as well, and stress has for the last decade become an increasing public health problem.” Please write distress or effects of stress/distress into this sentence, as stress in itself does not elicit a health problem.

Answer: This has been changed (line 39-40)

Page 2., Lines 52-55.: it seems that there is a missing part in this sentence.

Answer: This part has been deleted

Figure 2.: in this figure, one participants possible drop-out from the study was not illustrated between the „Eligible for treatment (n=154)” and the boxes of the intervention group and the control group.

Answer: There was a mistake in the box ‘Eligible for treatment’. The number of participants has been changed to 153.

Figure 2.: there is a mistake in this figure where TUA is written instead of TAU.

Answer: This has been changed.

Page 11-12., Lines 464-470.: „The reason for only 50% of the intervention group responding on the 6-month follow-up questionnaire could be that some of the participants were not used to using digital tools and therefore found it difficult to reply; the length of the questionnaire may have demotivated some of the participants; and the interest in the course and importance of completing the questionnaire may have paled 6 months after the intervention ended. Furthermore, the situation with COVID-19 has limited the contact with former participants since the healthcare centers have been closed for a period.” It would be illustrative if the Authors described the exact start date of the first intervention group and the date of the last 6-month follow-up examination of the project until the submission the manuscript.

Answer: The time for collection of data for this study has been included, Line 154-157.

Page 13., Lines 482-485.: „Approximately 50% of both groups of men had not completed a higher education, whereas the other half were relatively well-educated, since most of them had completed an intermediate or long higher education after finishing elementary school or high school.” This sentence seems to be a little judgmental, I think it would sound better to write it for example this way: Approximately 50% of both groups of men had not completed a higher education, whereas the other half had completed an intermediate or long higher education after finishing elementary school or high school.

Answer: This has been changed (line 405-408)

Page 15., Lines 520-522.: „Significant improvements were also shown on PSS with a reducing I stress symptoms by 3.63 (SD 5.13, p<0.001) and on PRS, increasing by 18.57 (SD 27.14, p<0.001).” I think it would be a possible way to correct this sentence by modifying its middle part like this:„reduction in stress symptoms”.

Answer: This has been changed (line 442-43)

Reviewer 2 Report

Overall the manuscript is good. I have some questions, clarifications and suggestions to help improve the overall presentation. I have attached my comments as a word document. 

Author Response

Thank you very much for your constructive comments and suggestions of improvements of our article. We hereby resubmit a revised edition of the article. Besides the changed made based on the comments of the reviewers, we have ourselves made larger reductions and changes in the text of the manuscript.

Attached you will find our answers to the comments of the reviewers.
